# Ecological and Molecular Interactions between Insects and Fungi

**DOI:** 10.3390/microorganisms10010096

**Published:** 2022-01-02

**Authors:** Rosario Nicoletti, Andrea Becchimanzi

**Affiliations:** 1Council for Agricultural Research and Economics, Research Centre for Olive, Fruit and Citrus Crops, 81100 Caserta, Italy; 2Department of Agricultural Sciences, University of Naples Federico II, 80055 Portici, Italy; andrea.becchimanzi@unina.it

**Keywords:** entomopathogens, mycophagy, mutualistic symbioses, mycobiome, insect immunity, crop protection

## Abstract

Insects and fungi represent two of the most widespread groupings of organisms in nature, occurring in every kind of ecological context and impacting agriculture and other human activities in various ways. Moreover, they can be observed to reciprocally interact, establishing a wide range of symbiotic relationships, from mutualism to antagonism. The outcome of these relationships can in turn affect the extent at which species of both organisms can exert their noxious effects, as well as the management practices which are to be adopted to counter them. In conjunction with the launch of a Special Issue of Microorganisms with the same title, this article offers a general overview of the manifold aspects related to such interactions from the perspective of implementing our capacity to regulate them in a direction more favorable for the environment, crop production and human health.

## 1. Introduction

Fungi and insects share common traits: they both possess chitin-based exteriors, both are heterotrophic and both can be detrimental to plants, representing the biological adversities with the highest impact on crops, in terms of both damage and necessity to use chemicals. Many cases of direct trophic relationships are known where fungi are exploited as a feeding resource by mycophagous insects, while in turn many fungi are entomopathogenic and make use of insects as a nutrient substrate. However, coevolution led to a continuum of interactions between these organisms which, far from being merely antagonistic, may also involve mutualistic relationships. The increasing awareness of the existence of direct and indirect ecological interactions has stimulated the consideration of insects and fungi associated with crops following the paradigm of integrated pest management, not only in view of combining control practices to contrast their noxious effects but also, and especially, to exploit their symbiotic relationships in such a way that they can result in a beneficial outcome for plants. Studies displaying how fungi can play a positive role, by directly affecting insect pest development or inducing plant resistance and defense reactions, have in turn stimulated insights on interactions going beyond these basic effects, which involve more strict developmental relationships requiring further elucidation.

## 2. Mutualistic Interactions

Insects and fungi are widespread in many environments where they have had the opportunity to interact for million years, reciprocally influencing in various ways. Extraordinary proof of this long coevolution is the lateral transfer of genes from fungi which underlies carotenoid production in aphids [1]. Further striking evidence of these ancient connections is represented by the capacity of many species of ants, termites and some Coleoptera, such as the ambrosia beetles and the ship-timber beetles, to cultivate fungi in their nests as the main food source [2]. As a result of these mutualistic interactions, in many insect groups external cuticular modifications have arisen to house fungal symbionts, such as mycangia in beetle—fungus symbioses [3]. With reference to nutritional aspects, fungivory is considered as the ancestral feeding habit of gall midges (Diptera, Cecidomyidae) for over 100 million years. In fact, most of the species in this family are believed to introduce a fungal symbiont providing support in both nutritional and protective terms in the larval chamber which is established within hypertrophic or modified plant tissues [4,5].

Besides cases where fungi directly represent the food source, many associations are known where fungi are hosted inside the insect’s gut, being essential for the digestion of nutritional challenging substrates, with reference to either a dominant cellulose content, or the need to neutralize toxic allelochemicals accumulated by plants in their tissues [6,7]. In other cases, the function of fungi is fundamental to help insects assuming some essential substances which they are unable to obtain from their specialized feeding source. Yeasts are particularly known to be involved in symbioses established in the digestive trait [8]. A unique kind of mutualism characterizes the association between *Yarrowia* yeasts (Saccharomycotina, Dipodascaceae) and the carrion burying beetles (Coleoptera, Silphidae, Nicrophorinae), where the yeasts from the beetle’s gut microbiota spread in the form of a biofilm over the carcass, protecting it from the proliferation of other microbial decomposers and promoting optimal larval growth [9]. Besides ‘true’ yeasts, these microfungal associates are taxonomically heterogeneous, including species in the Pezizomycotina, such as *Symbiotaphrina* spp. known as anobiid beetle symbionts [10], and the Basidiomycetes, such as *Rhodotorula* and *Cryptococcus* spp. reported in association with *Dactylopius* scales (Heteroptera, Coccidae) [11]. Quite interestingly, some species, such as the endosymbionts of planthoppers (Heteroptera, Delphacidae), are phylogenetically related to the Cordycipitaceae, a Sordariomycetes family including some of the best known entomopathogenic fungi [12]. Again, these ‘digestive’ associations are presumed to be ancestral, and in the case of the trichomycetes symbionts (*Smittium* spp.) (Harpellomycetes, Legeriomycetaceae) inhabiting the guts of many species of aquatic Diptera, it has been estimated that they were established during the origin of complete metamorphosis in these insects, around 300 million years ago [13]. In all these examples of mutualism, the fungal symbiont in turn has the evident advantage of being able to dwell in a specific micro-habitat and of increased dispersion opportunities, which eventually help it to colonize the surrounding environment and to spread over higher distances.

## 3. Host–Pathogen Interactions

On the other hand, most, if not all, insect species can be infected by obligate or facultative entomopathogenic fungi which exploit them as the only or prevalent nutrient source. The coevolutionary arms race between fungal entomopathogens and their hosts led to the diversification of sophisticated strategies to counter insect immune and behavioral defenses. Proteolytic enzymes and toxins produced by parasitic fungi during infection interfere with the host immune system by altering the cellular and humoral immune response [14,15], while the export of small silencing RNAs interferes with the expression of the host’s immune genes [16]. Rather than merely being opportunistic pathogens, *Aspergillus* spp. (Eurotiomycetes, Aspergillaceae) could have a regulatory impact on the immune system of honeybees through the production of phenoloxidase inhibitors, which interfere with the melanization response of insects [17,18]. Oosporein, a dibenzoquinone toxin secreted by *Beauveria bassiana* (Sordariomycetes, Cordycipitaceae), down-regulates the immune responses in mosquitoes’ midgut, causing dysbiosis and then septicemia after translocation of bacteria from the gut to the hemocoel [19]. However, only few studies have examined the effects of insect microbiota on host fitness and immunity in response to fungal pathogens.

The intimate interaction continuing through the ages has shaped entomopathogen fitness in such a way that they are sometimes able to induce behavioral responses by the susceptible insects [20,21]. In this respect, a nice example is represented by *Ophiocordyceps unilateralis* (Sordariomycetes, Ophiocordycipitaceae), a specialized parasite of ants in the tribe Camponotini (Hymenoptera, Formicidae), which just before killing its hosts, induces them to bite the underside of leaves close to the soil. Fixed in this position, corpses remain exposed to temperature and humidity conditions which are more favorable for the formation of larger fruiting bodies, consequently allowing the spread of a higher number of spores [22]. Behavioral response to fungal entomopathogens is also an important part of the defense strategies adopted by social insects, which are vigilant to rapidly detect cues of fungal pathogens, avoid direct contact with contaminated individuals, clean the body surface of nestmates by allogrooming, sanitize the nest with antimicrobials and remove dead individuals, reducing the probability of epizootic spread [23]. Volatile organic compounds (VOCs) produced by social insects and fungi are important communication cues which influence insect behavior and pathogen proliferation. For example, the volatile (2)-b-elemene produced by soldiers of the Japanese termite *Reticulitermes speratus* (Blattodea, Rhinotermitidae) is active as a pheromone and a fungistatic agent against *B. bassiana* and *Metarhizium anisopliae* (Sordariomycetes, Clavicipitaceae) [24]. Considering that fungal volatiles also influence the metabolic activity of bacteria [25,26], their effects on gut microbiota in insects, following the holobiont concept, deserve further investigations. VOCs play an even more important role in multitrophic interactions, allowing inter-kingdom communication between indirectly related communities.

## 4. Plant-Mediated Interactions

The role of fungi is essential in controlling insect populations and preserving homeostasis in the ecosystems, in such a way that they tend to establish mutualistic associations with the host plants by colonizing their tissues as endophytes [27,28]. In the context of the actions addressed at reducing the use of pesticides in agriculture, these properties are increasingly considered for their applicative perspectives in integrated crop management [29].

Plants can be considered as an active bridge between below and above ground biocoenoses. Root-colonizing and endophytic fungi interact with herbivore insects by modulating plant defense and by stimulating the production of plant VOCs which attract the natural antagonists of pests [30,31,32]. Colonization by *Trichoderma* spp. (Sordariomycetes, Hypocreaceae) induces the systemic defense response of plants against aphids [30,32,33] and Lepidoptera [32,34], and attracts both parasitoids [35] and predators [36]. Endophytic *B. bassiana* can control different species of aphids on a series of crops [37,38,39] and negatively affects the fitness of several Lepidoptera species [40,41,42,43,44], even if further investigations are required for more thoroughly understanding the mechanism of action. While many studies focus on herbaceous plants, little is known about multitrophic interactions which occur in and around the tree holobiont, including the endophytic and epiphytic mycobiota and how these can impact populations of forest pests [45].

Plant-mediated interactions between fungi and insects can also be mutualistic. Indeed, flower organs and nectar are commonly inhabited by yeasts (e.g., *Metschnikowia* spp., *Cryptococcus* spp., *Aureobasidium pullulans*) which have a significant impact on nectar scent, the foraging behavior of pollinators and parasitoid attraction [46,47]. The consumption of nectar colonized by yeasts has been shown to improve bee fitness, probably due to the increase in prebiotic hetero-oligosaccharides; however, the effects largely depend on the yeast species [47]. Analogously to their role as plant pollinators, insects may favor fungal sexual reproduction by carrying spermatia (gametes). The rust fungus *Puccinia monoica* (Pucciniomycetes, Pucciniaceae) produces spermatogonia in brilliant yellow pseudoflowers on its host plants (*Arabis* species) mimicking true flowers of unrelated species in shape, size, color and nectar production; these pseudoflowers attract insects which fertilize the receptive hyphae to form aecia [48].

## 5. Conclusions

As concisely outlined in the previous sections, the available literature provides valuable examples of how interactions between fungi and insects can shape rich and strikingly complex multitrophic systems in both natural and agricultural contexts. Many recent omics studies are focused on insect mycobiota, revealing an astonishing diversity of species and metabolic potentials, and introducing the opportunity for a practical exploitation in pest management. New methods to be adopted in this never-ending struggle may be conceived, based on the disruption of mycobiota associated with digestive functions or interferences in the completion of the pests’ life cycle. From an applicative perspective, suppressing the insect immune response using targeted gene silencing technologies, such as RNA interference, seems a promising way to enhance the efficacy of entomopathogenic fungi as biocontrol agents. Moreover, promising applications can be realized through the modulation of the tritrophic interaction with plants, particularly with reference to all aspects mediated by the various kinds of chemical messages encoded by fungal VOCs. Finally, possible applications in human medicine of the valuable results achieved in the study of insect–fungi relationships and the intimate processes underlying infection and immunity in insects can be expected based on the appreciation that defense mechanisms of these arthropods share many fundamental characteristics with the immune system of vertebrates.

## Data Availability

Not applicable.

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
