# Peer review of "Ecological and Molecular Interactions between Insects and Fungi"

_microorganisms, 2022, doi:10.3390/microorganisms10010096_

Round 1

Reviewer 1 Report

This review is well written and easy to read. However I often had the feeling that statements about the evolution of the interaction between insects and fungi were not backed up enough/detailed enough. Since this review explores interactions, I think this is an important point to address. An other important aspect that is missing for me is “the perspective of implementing our capacity to regulate them in a direction more favorable to the environment, crop production and human health.”. Could the authors provide any suggestions or examples of implements ? I believe these aspects can be concisely added t the manuscript and improve its value.

Lines 35-36, this sentence needs to be corrected.

Line 62, yeast occurring in the digestive trait does not make sense.

Lines 72-73: I think this sentence needs to be developed. Most entomopathogenic fungi are actually decomposers and opportunistic pathogens. It would be nice to give more detail on the evolutionary history of this symbiosis relative to other Cordyceps sp., especially what are the clues indicating symbiosis is the ancestral state ?

The same in the case of lines 85 to 87, what is the evidence for a coevolutionary arms race ? Are there any phylogenetic data ? Simply showing that there are one one side virulence factors and on the other immune effectors is not enough to conclude to a coevolutionary arms race. This sentence also needs to be developed. Immune effectors could have evolved against other classes of microorganisms, and virulence factors can have other roles than during parasitism. Oosporein for example is readily produced outside of the insect body.

Lines 122-123: “… is essential in controlling insect populations in order to preserve…” suggests fungi have the purpose of preserving a balance in the ecosystem, please correct.

Lines 131-132: does the colonization by this fungus affect plant fitness apart from attracting antagonists of insect pests ?

Line 145: is the mechanisms of this fitness improvement known ?

I hope my comments help,

Author Response

This review is well written and easy to read. However I often had the feeling that statements about the evolution of the interaction between insects and fungi were not backed up enough/detailed enough. Since this review explores interactions, I think this is an important point to address. An other important aspect that is missing for me is “the perspective of implementing our capacity to regulate them in a direction more favorable to the environment, crop production and human health.”. Could the authors provide any suggestions or examples of implements ? I believe these aspects can be concisely added t the manuscript and improve its value.

Thank you for your positive comments and help in reviewing the manuscript. However, we specify that it was conceived as an introductory paper to the special issue 'Ecological and Molecular Interactions between Insects and Fungi', hence we deliberately did not go into details for many aspects.

Lines 35-36, this sentence needs to be corrected.

Corrected.

Line 62, yeast occurring in the digestive trait does not make sense.

We modified this sentence.

Lines 72-73: I think this sentence needs to be developed. Most entomopathogenic fungi are actually decomposers and opportunistic pathogens. It would be nice to give more detail on the evolutionary history of this symbiosis relative to other Cordyceps sp., especially what are the clues indicating symbiosis is the ancestral state ?

The same in the case of lines 85 to 87, what is the evidence for a coevolutionary arms race ? Are there any phylogenetic data ? Simply showing that there are one one side virulence factors and on the other immune effectors is not enough to conclude to a coevolutionary arms race. This sentence also needs to be developed. Immune effectors could have evolved against other classes of microorganisms, and virulence factors can have other roles than during parasitism. Oosporein for example is readily produced outside of the insect body.

Indeed, these topics are fascinating. But again, we cannot go into further details on these, as well as many other interesting aspects which have been just introductorily treated.

Lines 122-123: “… is essential in controlling insect populations in order to preserve…” suggests fungi have the purpose of preserving a balance in the ecosystem, please correct.

This sentence has been corrected. However, our statement cannot be intended that fungi have the purpose of...., yet their action in the ecosystems contributes to a balance among plants and herbivorous organisms.

Lines 131-132: does the colonization by this fungus affect plant fitness apart from attracting antagonists of insect pests ?

The genus Trichoderma includes many species known as plant associates, which improve their host's fitness under several aspects. Literature in the field is quite extensive, and we just considered a few references giving a general outline of the interactions resulting in protection against insect pests.

Line 145: is the mechanisms of this fitness improvement known ?

This sentence has been modified.

Reviewer 2 Report

The reviewer has no major concern about this mini-review.

Review comment:

Manuscript title: Ecological and Molecular Interactions between Insects and Fungi

As the Perspective manuscript, the present one has made a brief but complete summarization for the ecological and molecular interaction between fungi and inset. The content summarized here, to some degree is helpful and inspiring for understanding the interaction between fungi and vertebrate as the authors mentioned in the work.

This Perspective note work also highlights the importance of further dissecting the interplay between fungi and insect.

The manuscript is written clearly, there is a good flow for reading, the reviewer does not have major concern about the manuscript.

Author Response

Thank you for your positive comments